



# Errors in top-down estimates of emissions using a known source

Wayne M. Angevine[1,2*], Jeff Peischl[1,2], Alice Crawford[3], Christopher P. Loughner[3,4], Ilana B. Pollack[5], and Chelsea R. Thompson[1,2]

[1]Cooperative Institute for Research in Environmental Sciences (CIRES), University of Colorado, Boulder, Colorado, USA
[2]NOAA Chemical Sciences Laboratory, Boulder, Colorado USA
[3]Atmospheric Sciences Modeling Division, Air Resources Laboratory, NOAA, College Park, MD, USA
[4]Cooperative Institute for Satellite Earth System Studies (CISESS) / Earth System Science Interdisciplinary Center (ESSIC), University of Maryland, College Park, MD, USA
[5] Department of Atmospheric Science, Colorado State University, Ft. Collins, Colorado USA

*Correspondence to:* Wayne M. Angevine, CIRES / NOAA R/CSD4, 325 Broadway, Boulder, CO 80305 USA email: Wayne.M.Angevine@noaa.gov phone: 303-497-3747

**Abstract.** Air pollutant emissions estimates by top-down methods are subject to a variety of errors and uncertainties. This work uses a known source, a coal-fired power plant, to explore those errors. The known emissions amount and location remove two major types of error, facilitating understanding of other types. Biases and random errors are distinguished. A Lagrangian dispersion model (HYSPLIT) is run forward in time from the known source, and virtual measurements of the resulting tracer plume are compared to actual measurements from research aircraft. Four flights in different years are used to illustrate a variety of conditions. The measurements are analyzed by a mass-balance method, and the assumptions of that method are discussed. Some of those assumptions can be relaxed in analysis of the modeled plume, allowing testing of their validity. Meteorological fields to drive HYSPLIT are provided by the European Center for Medium Range Weather Forecasts Fifth Reanalysis (ERA5). A unique feature of this work is the use of an ensemble of meteorological fields intrinsic to ERA5. This analysis supports reasonably large (30-40%) uncertainties on top-down analyses.

## 1 Introduction

Emissions of air pollutants must be known for modeling of exposure and planning for compliance with concentration standards. Bottom-up and top-down methods are used to estimate emissions. Bottom-up methods combine activity data with emissions factors, essentially counting sources and multiplying by their individual emissions. This is the main method used to produce official inventories. Top-down methods use measurements of atmospheric concentrations to estimate emissions. Both types of methods are subject to substantial uncertainty, and often disagree (e.g. (Hsu et al., 2010)).

The main purpose of this work is to evaluate some of the errors and uncertainties in top-down methods, while controlling other sources of uncertainty. Evaluating the errors and uncertainties of top-down methods is difficult. Not only the emissions amounts, but their location, distribution, and timing are often unknown. Attempts to constrain all these matters simultaneously result in grossly under-determined systems. Most often the under-determination is dealt with by employing Bayesian statistical methods, which introduce further errors and uncertainties, and remove the desired independence of the top-down and bottom-up methods.

In this work, we start with an emission source known in quantity, timing, and location. That source is the Martin Lake coal-fired power plant in eastern Texas. It is located in reasonably simple, flat terrain. Stack emissions of several gasses are measured by Continuous Emissions Monitoring Systems (CEMS). Concentration measurements from aircraft are used to estimate emissions by mass balance, and to compare with modeled concentrations. Here we use sulfur dioxide ($SO_2$), nitrogen oxides ($NO_y$), and carbon dioxide ($CO_2$). $SO_2$ is emphasized because its peak concentration measured in aircraft traverses is well-defined above the regional background. $SO_2$ is lost to surfaces and converted to sulfate aerosol, but that conversion is slow compared to the transport time of the main transects we use



here, which are usually 30-60 minutes downwind of the stack. CEMS data are also uncertain, but we assume that those uncertainties are small, i.e. <10% (Peischl et al., 2010), relative to the other uncertainties treated here. For the purposes of this paper, CEMS data are considered the "reality" with which other estimates are compared.

Emissions can be estimated from observations alone under certain non-trivial assumptions. Inverse modeling is used to allow relaxation of some of those assumptions. In this work, we replace some of the observations with (forward) modeled values, in varying combinations. This allows us to characterize and estimate errors arising from the models, and is a step toward understanding errors and uncertainties in inverse modeling.

In the mass balance method the object is to determine the amount of mass flowing through a plane downwind of a source. The mass flow rate through this plane is used as an estimate for the emission rate from the source. Sources of error are (1) error in determining the mass flow rate through the plane; (2) the emission rate at the source may be different than the mass flow rate through the plane; and (3) pollutant may be lost to deposition and/or chemical transformation. Determining the mass flow rate through the plane is done by estimating the concentration at each point in the plane, multiplying by the area to get a linear mass density and then multiplying by the wind speed perpendicular to the plane to obtain a mass flow rate. Usually one or more transects is flown through a plume during presumably well mixed conditions. Thus the concentrations at a single height may be used as an estimate for concentrations from the ground to a well-defined mixing height. Errors may arise because the plume is not well mixed causing concentrations to vary significantly in the vertical direction, the planetary boundary layer height is not well known, or significant mass has been transported above the boundary layer. The other source of error is that the mass flow rate through the plane may be significantly different than the emission rate from the source. This may occur when winds are variable in speed and/or direction in time and/or space.

By using a model, we can virtually eliminate errors arising from (1). In the simulated world, simulated concentrations and model wind speeds at every point in the plane are known so there is no need to estimate a mixing height. In addition, the emission from the source is also known in the simulated world. Thus mismatch between the emission rate from the source and the mass flow rate from the analysis plane is only caused by variable winds and can be explored in detail. Uncertainties due to losses to deposition and/or transformation (3) are also eliminated in the simulations.

The mass balance method can also be applied to the simulated fields by using only simulated concentrations from a single transect and the model mixing height. The tracer profile is known exactly in the simulated world, but the mixing height may still not be well defined. Tracer profiles may not have a sharp cutoff in the vertical. Significant mass is often found above the boundary layer height specified to the simulation because the model is designed to allow transport above the boundary layer and also because the boundary layer height changes in space and time. It is difficult to tell, however, if this accurately reflects vertical mixing in the real world.

We attempt to distinguish between error and uncertainty within this work. Error is the difference between an analyzed value and the true value, in this case, of the emission rate. Uncertainty is an estimate of the error that we expect in the absence of knowledge of the truth. (Metrology, 2008) Error and uncertainty have systematic and random components. Systematic error is synonymous with bias, that is, persistent differences of one sign between reality and a result of analysis. The distinction is neither precise nor crisp, and terms are not always used carefully. Atmospheric measurements rarely have enough samples to reliably distinguish the two. Bias can be introduced by the use of methods or assumptions that most often move the result in one direction. For example, a low bias in a wind speed measurement will result in a low bias in the emissions estimated by mass balance. Random differences have many possible causes, one important cause being sampling uncertainty, that is, the difference between the mean of a quantity measured with a small number of samples and the true (ensemble) mean. The distinction is important for several reasons. Reporting a small uncertainty with possibly large unknown biases can lead to incorrect policy decisions. Bayesian analyses assume that the measurements and prior have zero mean error and (usually) Gaussian uncertainty, and the proper characterization of the error covariances is critical to a good result. From a practical point of view, repeated measurements can reduce random uncertainty but cannot reduce bias.

Errors and uncertainties in the meteorological fields (modeled or measured) used in analyses propagate directly into the result. Since we only measure one realization of the chaotic atmosphere, we never have an ensemble average (statistically speaking). Normally, only a single meteorological field is used. The numerical weather prediction community has been moving steadily toward producing ensemble output, that is, well-designed sets of multiple realizations. These can provide a rough method to distinguish between bias and random uncertainty. The difference





between the result produced from a control run and reality is an estimate of bias. So is the difference between an ensemble mean result and reality. The statistics of differences between results from all ensemble members are
105 estimates of random uncertainty. The quality of these estimates depends on the design and quality of the ensemble.

Meteorological quantities have been identified as major sources of error and uncertainty in emissions estimates. These can be divided into several classes. Wind direction errors displace the plume (or the source in an inverse analysis). Wind speed errors change the magnitude of the plume and its timing with respect to time-varying emissions. Mistaken
diagnosis of the mixing height affects the concentrations and may contribute to violation of the well-mixed assumption. Errors in the transport model are also important. Under- or over-estimation of horizontal dispersion changes the plume width. Discretization in either horizontal or vertical dimensions may add noise or uncertainty. Physical situations that are correctly handled in the models may still lead to errors in some analyses, for example temporal variation (unsteadiness) of wind can result in storage of pollutant that violates the assumption of steady wind in mass balance
analysis. This can be exacerbated if the winds are not updated often enough in the transport model.

## 2 Data

The Martin Lake power plant complex is located in fairly simple terrain in east Texas (32.260ºN, -94.570º). It has
120 three stacks that are each 452 ft (138 m) high. The stacks are spaced 100 m apart.

The measurements used here were taken by NOAA scientists aboard NOAA or NCAR research aircraft. Flights downwind of the Martin Lake power plant were made in four years (2000, 2006, 2013, and 2015). Dates are shown in Table 1. The flights were planned to intercept the plume at least once and usually several times. Downwind distances
were chosen to satisfy the conditions for mass balance analysis (see below), far enough downwind for the plume to be well-mixed through the boundary layer in the vertical but close enough for the concentration signal to be strong and to minimize chemical transformations.

For all four flights, $SO_2$ was measured using a modified, commercial, pulsed UV fluorescence instrument, a Thermo
Environmental Instruments, Inc., model 43S (Ryerson et al., 1998). The 1 Hz measurements have an estimated 1-sigma precision of ±0.3–1 ppbv and a 1-sigma uncertainty of ±10–12%, depending on year. NOy was measured by chemiluminescence after conversion to NO in a heated gold catalyst (Ryerson et al., 1999). The 1-Hz precision ranged from 0.015–0.4 ppbv, and uncertainty ranged from ±10–12%, depending on year. $CO_2$ was measured by infrared absorption using a LI-COR 6262 in 2000 and 2006 (Peischl et al., 2010), and a Picarro 1301-m in 2013 and 2015
(Peischl et al., 2012). The 1-Hz precision was at or below ±0.1 ppm for all flight years, and the measurement uncertainty was ±3% in 2000, and approximately ±0.15 ppm for the other flights.

Pursuant to federal regulation, commercial electric utility steam generating units with a capacity greater than 25 MW are required to monitor stack emissions and report them to the Environmental Protection Agency. The full capacity of
140 each of the three units at Martin Lake are greater than 700 MW, thus subject to reporting requirements. The relative accuracy of the $SO_2$, NOx, $CO_2$, and flow measurements are all required to be less than 10%. Therefore, we expect the uncertainty of a mass flow value or a ratio of two pollutant concentrations to be less than 14% after quadrature addition of the uncertainties. The ratio of pollutants was verified by top-down measurements of 11 Texas power plants, including Martin Lake, in a 2006 study (Peischl et al., 2010), but a determination of the mass flux has not.

## 3 Methods

Mass balance is a time-honored method of converting concentration measurements to emissions with minimal reference to models (e.g. (White et al., 1976; Trainer et al., 1995)). In the simplest cases, such as those presented
below, it assumes that a plume is well-mixed in the vertical to a well-defined height and that the wind speed and direction are steady. A robust estimate of the background (concentration not attributable to the source of interest) is required. At least one transect across the plume is made at a height well within the mixed layer. The mixed layer





height is usually determined from profiles at the ends of some transects. More elaborate methods involve multiple downwind transects, often accompanied by 2-D interpolation of downwind measurements (Mays et al., 2009).
Detailed analysis of related methods in the context of column measurements is given by Varon et al. (2018).

Fully model-based retrievals of emissions from concentration measurements are usually done with a Bayesian analysis in order to overcome the under-constrained nature of the problem. A Lagrangian dispersion model is run backwards in time from the sites of measurements to produce footprints, which are convolved with a prior inventory. The
differences between modeled and measured concentrations are then optimized by a mathematical algorithm. Weights (error covariances) are applied to the measurements and the prior, expressing relative confidence in their correctness. Bayesian analysis requires assumptions about the PDF of errors (usually Gaussian, always with zero mean).

Here we use hybrid or intermediate methods to examine the consequences of different classes of error. The
meteorological model (reanalysis) produces full fields of wind speed and direction and of mixing height. The Lagrangian transport model then produces a full four-dimensional view of the plume. From these we can calculate the concentrations at the aircraft location for direct comparison and for standard mass balance analysis, as in (Karion et al., 2019). We can also calculate other estimates by considering different heights within the plume, a range of heights, or the whole plume. We can use measured or modeled mixing heights, and measured or modeled winds. We can choose
whether to be sensitive to horizontal displacement of the plume.

The ECMWF fifth-generation reanalysis (ERA5) [Copernicus Climate Change Service (C3S) (2017): ERA5: Fifth generation of ECMWF atmospheric reanalyses of the global climate . Copernicus Climate Change Service Climate Data Store (CDS), accessed March 2019. https://cds.climate.copernicus.eu/cdsapp#!/home] is used to provide
meteorological fields for this study. It consists of a control (high resolution) run and 10 ensemble members. The control run is available hourly on a 0.25x0.25 degree latitude-longitude grid, with 37 pressure levels. The ensemble members are available every 3 hours on a coarser 0.5x0.5 degree grid.

HYSPLIT version 944 was used in this study. HYSPLIT is a Lagrangian atmospheric transport and dispersion model
developed by the National Oceanic and Atmospheric Administration's Air Resources Laboratory (NOAA ARL) (Stein et al., 2015). Dispersion of a material is simulated by a number of computational particles which represent a specified amount of mass of material. The computational particles are advected by the wind field and dispersed by a turbulent component which is calculated by the model from meteorological data fields.

HYSPLIT provides a variety of options to optimize for different situations. We used the default methods for determining vertical velocity variances (Kantha-Clayson), and wind and temperature profiles for computing the boundary layer stability. These are the same settings used by Karion et al. (2019). The mixed layer depth was taken from the ERA5 input, which is also the default method. HYSPLIT was modified to include the Stochastic Time-Inverted Lagrangian Transport (STILT) dispersion algorithm, which was employed in this study. The STILT dispersion
algorithm incorporates the (Thomson et al., 1997) reflection/transmission scheme for Gaussian turbulence that preserves well mixed distributions for particles moving vertically across interfaces between grid cells as described in (Lin et al., 2003).

A tracer was emitted from the location of the Martin Lake stacks (32.26°N, -94.57°) at 100 m with the rate given by the
hourly CEMS data for $SO_2$. Heat content of $8.5 \times 10^7$ W was specified for all simulations and used in the plume rise calculation.

Concentrations from HYSPLIT were output on a horizontal grid with increments of 0.011° latitude by 0.009° longitude, spanning 0.6° x 0.8°. Vertical levels (25) were spaced every 100 m up to 2000 m AGL and then every 200 m up to
3000 m AGL. HYSPLIT concentrations were output as averages between each defined level, so for example the first level in the output is the average between 0 and 100 m AGL. The hourly output represents the average over each hour. No deposition was used, and the particles were defined as entirely passive. Ten thousand particles per hour were used to produce a sufficiently smooth concentration field.



## 4 Results

We first examine aircraft observations from the flight on 25 June 2013, which took place in good conditions for mass-balance analysis as described above. The plume from the Martin Lake power plant was intercepted by the aircraft four times, as shown in figure 1. Figure 2 shows the time series of the observations and the plumes simulated using the control meteorology and the ten ensemble members. The simulated plume is well-aligned with the observations in the first transect, but displaced in the other three. Note that the third and fourth transects were flown in opposite directions. All the simulated plumes are weaker and wider than observed. There is little visually apparent spread in the ensemble, all the members produce plumes of similar location, magnitude, and width. As figure 3 shows, however, the integrated amount of $SO_2$ in the plumes does vary.

The next step is to compute the emission rate represented by each transect. This is done by mass balance using observed and modeled values separately. The observational analysis uses the observed mixing ratios, mixing height, and wind speed. Simulated plumes from the control run are analyzed with the simulated wind speed and mixing height. A full-plane integration of the simulated plume is also conducted, and described below. Thus we have a total of three emission rate estimates from the control simulations for each transect to compare with the CEMS measurement. Figure 4 shows these results. The CEMS data show that the aircraft transects all measured emissions that came out of the stack during a plateau of relatively constant emissions. In the lower panel of fig. 4, the emission rates are presented as ratios to the CEMS data for each species. By this means, the three measured species constitute separate estimates of the same emissions. The single simulated tracer, labeled "$SO_2$ sim", serves for all species.

Transect 2, at the shortest downwind distance, best meets the mass balance assumptions. The error bar shows the estimated uncertainty (±30%, one standard deviation) of the observed mass balance estimate. The estimate itself is nearly perfect, falling just below the unity ratio line. The simulated mass balance estimate is about 20% high, within the error bar. $NO_y$ and $CO_2$ observed estimates also fall within the estimated error (not shown). For the other transects, the $SO_2$ observed falls consistently below the unity ratio line, although only the farthest transect falls outside the error bar. The 30% error estimate is only shown for transect 2, but is the same for the other transects and species, and is described in more detail in the Discussion section below. $NO_y$ and $CO_2$ estimates are scattered, but within the 30% uncertainty estimate. The simulated tracer is nearly perfect for transects 1, 3, and 4, the transects at 45-52 km downwind.

The set of emission rate estimates resulting from the ensemble meteorology are shown in figure 4 with red + marks. For the closest transect (2), the ensemble spread is about 15%, less than the estimate of uncertainty for the observations. The ensemble spread contains the control simulation value, but does not span the unity line. The ensemble estimates for the farther transects have greater spread, although still somewhat less than the observation estimate, and all span both the control simulation values and unity ratio.

The above mass balance analyses are not sensitive to errors in the width or displacement of the plume, since they involve integrating across each plume regardless of its exact location or width. We can also do an analysis that is insensitive to mixing height by integrating the simulated plume in a vertical plane along the flight transect, which we call the full-plane integration. Error in the emission estimate from this method would arise only from deviations from the assumption of a steady mass flow rate through the plane. For transect 2 on 25 June 2013, the full-plane integration using the control simulation gives an emission rate of 7500 kg h$^{-1}$ (table 1) compared to the CEMS value of 6780. This is a bias of 11% (all percentage values are with respect to the CEMS value). The full-plane integration using the ensemble meteorology produces estimates ranging from 6900 to 7500 kg h$^{-1}$, with mean 7130 and median 7140 kg h$^{-1}$, a bias of 5%. The ensemble range of the full-plane estimates does not include the CEMS value or the observation-based mass balance value, but does include the control simulation mass balance value. The ensemble spread is rather small (8%). The simulated plumes are not perfectly well-mixed, with higher concentrations in the lower BL (figure 5).

The question of mixing height deserves further exploration. Figure 5 shows vertical cross sections in the along-wind direction of the tracer mixing ratios simulated by HYSPLIT with the control meteorology and each of the ten ensemble members. An observed value of mixing height was subjectively determined from potential temperature and water vapor profiles flown at the ends of the transects, shown as an o mark in each of the subplots. A simulated mixing height is estimated from the tracer mixing ratio profiles. It is the height at which the mixing ratio first falls below 50%





of its value in the middle height range. It differs for each ensemble member. Another possible source of uncertainty in the mixing height is shallow cumulus clouds, which were present on all four flight days.

Flux estimates for $SO_2$ for all four transects on 25 June 2013 are given in numerical form in table 1. The estimates from observations are within ±14% for three of the transects (2, 3, and 4). Simulated values from the control run are 5-19% high for transects 1, 2, and 4, and 14% low for transect 3. The observational estimate for transect 1 has a substantial low bias (34%), for which we do not have a convincing explanation.

Seeing that the observations and control simulations produce small biases for transects 2, 3, and 4, we now examine the ensemble simulations. These are shown in figure 4, and numbers are given in table 1. The ensemble does not span reality for transect 2, but does cover the control estimate. The ensemble spread is 12% for transect 2 and 25-36% for the other transects.

Another flight with several transects took place on 16 September 2006 (figure 6). Transect 2 has the best match to the mass balance assumptions. The emission rate analysis from observations has a high bias of 25%, while the estimate from the control simulation is biased 5% low. The observational estimates for $CO_2$ and $NO_y$ are close to their respective CEMS values (table 2). The full-plane estimate agrees closely with the observational estimate for $SO_2$ (table 1). The ensemble estimates have a spread of 36%, cover the CEMS value, and are nearly centered around the control simulation value. As for the other transects, the closest transect gives a high-biased observational estimate. The transects farther downwind all produce low-biased observational and simulated estimates, except for $CO_2$ and $NO_y$ at the farthest distance (not shown). The emissions as measured by CEMS are increasing during the span of time when the plume was emitted, adding substantial uncertainty to the comparison. Examining the vertical cross-sections of the plume (figure 7) we see that the plume is approximately well-mixed at the latitude of transect 2 (32.5°N) in all but one ensemble member (em8), but not well-mixed at 32.4°N, the latitude of transect 1.

On 3 September 2000, only one transect is usable (table 1, figure 8). The observational analysis produces an emission rate within 4% of CEMS. The control simulation overestimates by 19%. The ensemble estimates have substantial spread (46%), which covers the values from the observations and the control simulation. The full-plane integration has a 44% high bias. Some of the difficulty in the simulations is due to unrealistically large mixing heights in ERA5. HYSPLIT cannot produce a well-mixed plume in these conditions. Agreement between the observational analysis and CEMS reflects a reasonable mixing height estimate but may involve some element of good luck. We do not know whether the real plume was well-mixed. Potential temperature profiles (not shown) before the transect show relatively shallow mixing heights consistent with the manual estimate. Observed profiles after the transect show a deep BL ~2500 m AGL, although not as deep as in ERA5 (3000-3600 m). This is an indication that the mixing height was changing during the time of the observations. Both profiles are at some distance from the plume location, so their applicability is questionable. The emissions measured by CEMS are increasing substantially around the time the plume was emitted, which adds to the uncertainty.

The flight on 25 April 2015 took place after the $SO_2$ emissions of the power plant had been substantially reduced by scrubbing, so the $SO_2$ plume was much weaker. Analysis of the observations requires estimating the background, and uncertainty in that estimate is more important when the plume is weaker. The observational analysis produces a flux estimate biased 40% low; the control simulation is biased only 10% high (table 1, figure 8). The ensemble estimates have large spreads, more than 100%. The spread is due mostly to a single high member. We cannot justify removing that member, which is not obviously wrong. Wind speed is biased low and mixing height is biased high in the simulations (table 3). Several members produce concentration profiles that are not well-mixed at the observation plane (not shown). Potential temperature profiles before and after the transect (not shown) are consistent with the respective model and observed BLHs.

## 5 Discussion

In this section, we use the collection of emissions estimates described above to explore the uncertainty of such estimates. Several approaches are used:

1. Manual uncertainty estimates for the observational mass balance



2.  Errors from the observational mass balance for the four flights, including multiple species
3.  Errors from the simulated mass balance for the four flights
4.  Errors from the simulated full-plane integrations for the four flights
5.  Errors from simulated mass balance using the meteorological ensemble

Uncertainty estimates for mass balance calculations from observations alone are described in detail by (Peischl et al., 2015) for the Haynesville shale area, which includes the Martin Lake power plant, using the same flight data used here for 25 June 2013. They estimate uncertainties of 300m (about 20%) for mixing height and about 28% for wind speed.
These are the dominant uncertainties in their presentation. Combined, all sources yield flux uncertainties (for methane from the oil and gas field) of about 35%. We refer to these as the "manual" uncertainty estimates (list item 1 above). We estimate the total uncertainty in the mass balance emission rate by summing in quadrature the following sources of uncertainty: The accuracy of the gas species measurement ($\pm0.10$ ppmv + 3% in 2000, $\pm0.15$ ppmv otherwise for $CO_2$; $\pm10\%$ for $SO_2$, $\pm12\%$ for $NO_y$); the background determination of the gas species measurement ($\pm20\%$ for $CO_2$; $\pm5\%$ for
$SO_2$; $\pm10\%$ for $NO_y$), the boundary layer depth ($\pm200$ m), and the wind speed ($\pm1$ m/s). This results in a total 1-sigma uncertainty between $\pm22\%$ and $\pm32\%$ for the emission estimates presented here. For clarity of presentation in figures 4 and 6, this is represented by a $\pm30\%$ error bar.

Of the transects in table 1, many have large biases, but these problematic transects are either far from the source or
very close. We chose one "primary" transect for each day, flown at a roughly optimal distance from the source. The primary transects are highlighted in Table 1, and emissions of all three species are included in Table 2. Because the simulation uses a passive tracer, values scaled to the CEMS emission rates are applicable to all three species. Of these four transects, three have observational estimates within the manually estimated uncertainty for all three species. The outlying observational estimate, for 25 April 2015, is subject to uncertainty in the background, which may be
underestimated, and which does not affect the simulations. The $CO_2$ observational estimate is within the uncertainty on this day. It is not clear how to rigorously combine uncertainties for different flight days and species, so we do not present a single number for this collection (list item 2 above). An uncertainty estimate of less than 23% would leave four outliers rather than two. We therefore choose an estimate of approximately 25% to avoid being too optimistic.

All four estimates for $SO_2$ from the primary transects using the control simulation are within the manually estimated uncertainty (Table 1 and figure 8). Again, no rigorous combined uncertainty can be computed (list item 3 above) but anything less than 20% would be too optimistic, leaving out two of the four estimates. The full-plane integration (list item 4) eliminates one source of uncertainty, the estimation of mixing height. Somewhat counter to our expectation, however, this reduces the error in only two of the four cases. The error in mixing height must be offsetting an error due
to unsteady winds in the other two cases. The collection suggests a lower limit of 30% or so on the uncertainty.

Estimating probabilities or confidence intervals from small ensembles is challenging (Leutbecher, 2019; Wilks, 2011). Too few members will generally produce too small a spread. Ensemble spread is often expressed as a standard deviation, but if each member is equally likely, the range is a better measure. Even so, most ensembles have too little
spread, and do not encompass the true value. If biases occur, for example because of model errors common to all members, the spread probably should not encompass the true value. In other words, increased random spread is not an adequate substitute for diagnosis and correction of biases. Bias correction as part of ensemble calibration should be used. However, sufficient and reliable observations are often unavailable with which to correct biases, as is the case here.

The example of 25 June 2013 transect 2 is instructive (table 1). Emission rates from the ensemble range 12%, but do not cover the true value. If we call the ensemble range the uncertainty of the rate estimate, it would be too small (overconfident). If we used the ensemble standard deviation, the uncertainty estimate would be even smaller, about 5%, which is clearly unreasonable. For the same transect (25 June 2013 transect 2) the rate calculated from
observations is 8% low relative to reality, and the rate calculated from the control simulation is 19% high. Both estimates fall within the estimated uncertainty of the manual analysis.

Taking the other primary transects in order starting with the worst control flux estimate, for 3 Sept. 2000 we have a 19% high bias. The ensemble range is 46%. These differences suggest a somewhat larger uncertainty than the manual
estimate. Other clues to problems with this plume include very high BLH from ERA5 and a large difference between mixing heights estimated from the aircraft profiles before and after the transect. However, if mixing height were the only problem, the full plane integration would produce a correct estimate, since it is not sensitive to mixing height or to





the well-mixed assumption. The remaining source of error is the variability of the winds leading to storage of pollutant within the volume between the source and the measurement plane, which accounts for the rest of the bias (control) and spread (ensemble). The winds were light and variable during the night and early morning.

Transect 2 on 16 Sept. 2006 has emission rate estimates from the observations that are 25% high, and from the control simulation 5% low. The full plane integration returns a 22% high bias. The ensemble range is 36%, and covers reality. The other transects on this day are either too close for the plume to be well-mixed (transect 1) or too far away for the assumption that $SO_2$ is roughly conserved to be valid.

Transect 2 on 25 April 2015 has already been mentioned as suffering from background uncertainty. The control-based estimate is close to reality. The full plane integration is biased 28% low, still within the manually-estimated uncertainty. One outlying member produces the very large ensemble spreads. There is nothing to justify removing this member, which again emphasizes the difficulty of estimating spread from a small number of members.

Overall, the ensemble ranges give uncertainty estimates of 12-105% on different days (list item 5). This is information that cannot be obtained in any other way. We must caution, however, that the small range on 25 June 2013 is probably underestimated.

Mixing height and the well-mixed assumption are important uncertainties in the mass balance framework. This is not very surprising considering that the definition of a single mixing height involves several unsafe assumptions. For the boundary layer to be well-mixed, there must be a substantial surface buoyancy flux, a well-defined capping layer (not necessarily an inversion per se), and minimal change of advection with height. Boundary layers are commonly well-mixed in potential temperature but not in water vapor mixing ratio, because the entrainment fluxes at the top of the boundary layer are of opposite sign (warming for temperature, drying for water vapor). Additional conditions are required for a plume to be well-mixed, specifically there must be enough time (several boundary layer turnover times) between emission and measurement. It is worth noting that plumes that are not well-mixed can sometimes give reasonable mass balance estimates if the measurement transect is flown close to the middle of the boundary layer, where the measured value approximates the mean of the (roughly linear) vertical profile.

Wind speed is another important uncertainty. The usual assumption is that the wind speed at the measurement transect is representative for the entire plume transit. If the wind is light and therefore variable, or if the wind speed has changed substantially over the transit time, it is unclear what wind speed to use in the calculation. Sometimes modeled wind speeds are used (Karion et al., 2015). Looking at table 3, we see substantial biases in wind speed (low) and mixing height (high), the smallest biases occurring on 25 June 2013. These biases are with respect to the winds and mixing height observed on the aircraft, and therefore also include substantial uncertainties. The ensemble ranges can be compared with the manual estimates of uncertainty. For wind speed, the ensemble ranges are comparable to the manual estimates or smaller. For mixing height, the ensemble ranges are larger. Combining simulated and observed values, for example using simulated wind speeds with observed mixing heights, would incur large errors.

The methods used here are not sensitive to some common sources of error. None of these methods are sensitive to plume displacement caused by errors in wind direction, as long as the plume is fully covered by the transect. The full plane integration is not sensitive to the mixing height estimation or to violations of the well-mixed assumption. The fact that the full-plane esimate is farther from reality than the traditional mass balance estimate for three of the four primary transects suggests that compensating errors are present. Significant uncertainties remain in all methods.

This work was partly inspired by the study of Karion et al. (2019). The main emphasis of that work was on large (factor of 2 or greater) biases due primarily to errors in vertical mixing in the Lagrangian transport models. We should keep in mind, however, that the true emissions in that study are not known. The true vertical mixing is not known in either that study or this one. Karion et al. (2019) also reported large uncertainties (error bars) based on multiple flights and multiple models. Variability from a small ensemble of meteorology in one configuration (WRF2-FP) was 20-30%.

An earlier study by Angevine et al. (2014) used a six-member WRF ensemble driving forward runs of the Flexible Particle (FLEXPART) Lagrangian dispersion model to estimate uncertainty due to meteorology. A passive tracer representing national inventory emissions of carbon monoxide (CO) was transported by the models. That study found 30-40% spread in CO concentrations. The ensemble spread was insufficient to cover the errors in wind and temperature, as would be expected for a small ensemble using a single model framework.



## 6 Conclusions

Errors in top-down emissions estimates are substantial even under good conditions. Using a known source in forward runs driven by ensemble meteorology, we have shown errors ranging from a few percent to over 100%. No single robust estimate of either bias or random error can be derived from these results. Identification and removal of cases that are clearly bad reduces the error, but if the source is unknown, some bad cases cannot be clearly identified. Investigator judgement is vital. Using forward modeling and examining the structure of the plumes it produces can be helpful in identifying better or worse cases. Using an ensemble provides another dimension for diagnosis, since the spread of the ensemble results can indicate the possibility of outlying solutions that are not apparent in a single deterministic run.

The largest source of error is the vertical mixing, as was also shown by Karion et al. (2019). This is not simply a question of finding a correct mixing height, although that is a major problem. In several of the cases shown here, the plumes are not well-mixed, and the mixing height is therefore not well defined.

Wind speed is another important source of error (table 3). The wind speed on the aircraft transect may be the only source of data for comparison to the models. However, even if it compares perfectly, unsteadiness of the wind previous to the measurement can cause poorly-characterized errors.

Losses of pollutant to deposition or chemical transformation can be important for observations taken at longer downwind distances (greater than ~20 km), as seen in figures 4 and 6. These losses can be estimated if sufficient information is available, for example measurements of product species, but those estimates will necessarily introduce additional uncertainty.

These results are not sensitive to errors in plume location or width. In future work, we will explore the sensitivity of Bayesian inversion methods to those errors as well as to the kinds of errors shown here.

Our results are consistent with uncertainty estimates from rigorous analysis of observations, as performed by Peischl et al. (2015). The minimum emissions flux uncertainty that can be supported by these results is 30%. Under less than ideal conditions, errors can be much larger.

**Code and Data Availability**
Observational data used here are available from https://esrl.noaa.gov/csd/field.html. The HYSPLIT model and documenation are available from https://www.arl.noaa.gov/hysplit/hysplit/. HYSPLIT output used in this work is available at https://esrl.noaa.gov/csd/groups/csd4/modeldata/. The ERA5 analyses were generated using Copernicus Climate Change Service Information (2019). "Neither the European Commission nor ECMWF is responsible for any use that may be made of the Copernicus Information or Data it contains."

**Author Contributions**
WMA and JP designed the study. JP did the observational mass flux analysis including its uncertainties. WMA ran the HYSPLIT model with help from AC and CPL. WMA analyzed the model results and wrote the paper with input from the coauthors. IBP and CRT were involved in taking the measurements.

**Competing Interests**
The authors decare that they have no conflict of interest.

**Acknowledgements**
We are grateful for many enlightening discussions with Michael Trainer and Anna Karion. John Holloway, David Parrish, Steven Sjostedt, and Thomas Ryerson were involved in taking the measurements in one or more of the flights used here.



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





**Tables and Figures**

| Date | CEMS SO$_2$ rate (kg h$^{-1}$) | Mass balance SO$_2$ rate [bias] | Simulated mass balance SO$_2$ rate [bias] | Ensemble range SO$_2$ rate | Full plane integration SO$_2$ rate [bias] | Distance downwind (km) |
|---|---|---|---|---|---|---|
| 20130625 (transect 1) | 6380 | 4251 [-2129, -34%] | 6414 [34, 5.3%] | 5174-6708 (25%) | | 51 |
| **20130625 (transect 2)** | 6780 | 6727 [53, -7.8%] | 8091 [1311, 19%] | 7405-8323 (12%)* | 7500 [720, 10%] | 16 |
| 20130625 (transect 3) | 6780 | 5818 [-962, -14%] | 6688 [-92, -14%] | 5709-7752 (31%) | | 46 |
| 20130625 (transect 4) | 6780 | 6318 [-462, -6.8%] | 6903 [123, 18%] | 5533-7988 (36%) | | 46 |
| 20060916 (transect 1) | 7403 | 13227 [5824, 79%] | 10287 [2884, 40%] | 6501-10187 (46%) | | 16 |
| **20060916 (transect 2)** | 7403 | 9273 [1870, 25%] | 7045 [-358, -4.8%] | 5472-7923 (36%) | 9052 [1649, 22%] | 27 |
| 20060916 (transect 3) | 7403 | 2673 [-4730, -64%] | 6020 [-1383, -19%] | 4715-6658 (34%) | | 37 |
| 20060916 (transect 4) | 7403 | 3186 [-4217, -57%] | 5959 [-1444, -20%] | 4337-7319 (49%) | | 52 |
| 20060916 (transect 5) | 7403 | 3914 [-3489, -47%] | 6247 [-1156, -16%] | 4371-7136 (46%) | | 52 |
| 20060916 (transect 6) | 10404 | 7955 [-2449, -24%] | 5677 [-4727, -45%] | 4070-6461 (43%) | | 52 |
| **20000903 (transect 1)** | 9105 | 8773 [-332, -3.6%] | 10870 [1765, 19%] | 7696-12717 (46%) | 13154 [4049, 44%] | 10 |
| **20150425 (transect 2)** | 418 | 251 [-167, -40%] | 461 [43, 10%] | 350-893 (105%) | 299 [-119, -28%] | 36 |



Table 1: Values and ranges of $SO_2$ emission rates from several estimates as described in the text. Asterisk denotes ensemble ranges that do not include the true
(CEMS) value. The four primary transects identified for further analysis are in bold type, and full-plane values are provided only for those four primary
transects.





| Date | CEMS SO$_2$ rate (kg h$^{-1}$) | Mass balance SO$_2$ rate [bias] | CEMS NO$_x$ rate (kg h$^{-1}$) | Mass balance NO$_x$ rate [bias] | CEMS CO$_2$ rate (kg h$^{-1}$) | Mass balance CO$_2$ rate [bias] |
|---|---|---|---|---|---|---|
| 20130625 (transect 2) | 6780 | 6727 [-7.8%] | 1610 | 1782 [11%] | 2.51e6 | 2.39e6 [-4.7%] |
| 20060916 (transect 2) | 7403 | 9273 [25%] | 1950 | 1941 [-0.5%] | 2.55e6 | 2.25e6 [-12%] |
| 20000903 (transect 1) | 9105 | 8773 [-3.6%] | 3173 | 3077 [-3.0%] | 2.40e6 | 2.53e6 [5.4%] |
| 20150425 (transect 2) | 418 | 251 [-40%] | 398 | 132 [-67%] | 4.21e5 | 3.26e5 [-23%] |

Table 2: CEMS and observed emission rates for SO$_2$, NO$_y$, and CO$_2$ for the four primary transects. SO$_2$ values are
repeated from Table 1.





| Date | Wind speed mean (m s-1) [bias] | Wind speed ensemble range | Mixing height (m) [bias] | Mixing height ensemble range |
|---|---|---|---|---|
| 20130625 (transect 2) | 6.4 [-0.77, -11%] | 1.4 (21%) | 1900 [170, 9.8%] | 500 (26%) |
| 20060916 (transect 2) | 5.1 [-1.8, -26%] | 0.95 (19%) | 1617 [398, 33%] | 800 (44%) |
| 20000903 (transect 1) | 4.7 [-0.88, -16%] | 1.5 (32%) | 1900 [681, 56%] | 1300 (58%) |
| 20150425 (transect 2) | 5.5 [-1.7, -23%] | 1.3 (23%) | 1600 [614, 62%] | 600 (43%) |

Table 3: Values and ranges of wind speed and mixing height in the simulations, and their biases with respect to the
observed values.





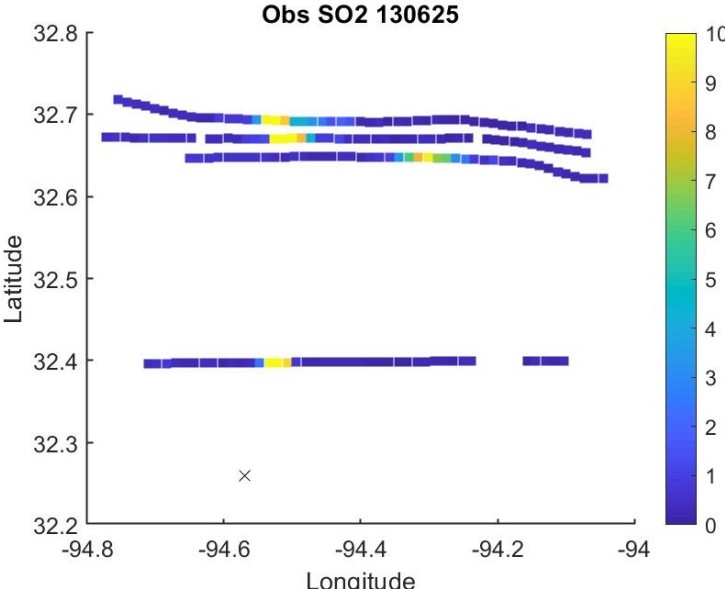

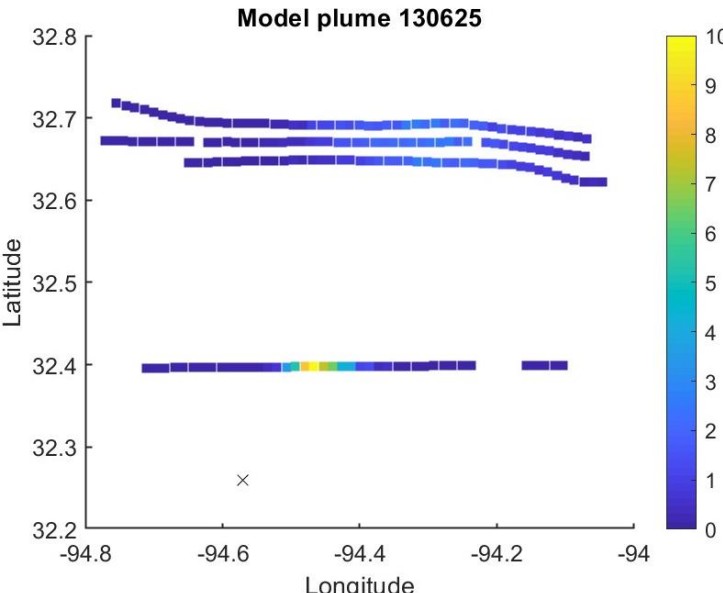

Figure 1: SO₂ mixing ratios (ppbv) observed along four transects by the aircraft (top) and modeled at the same
locations (bottom) in the control run on 25 June 2013. The x marks the power plant location.

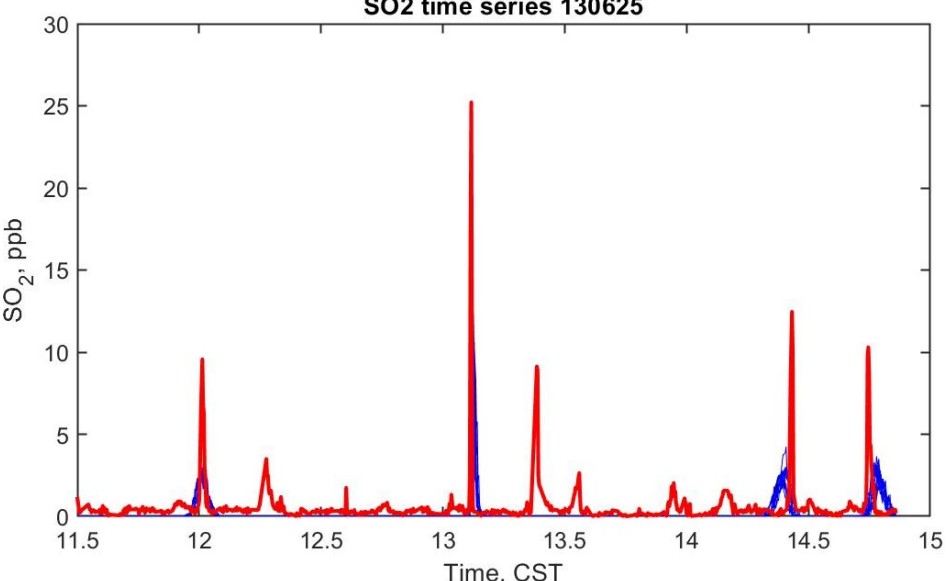

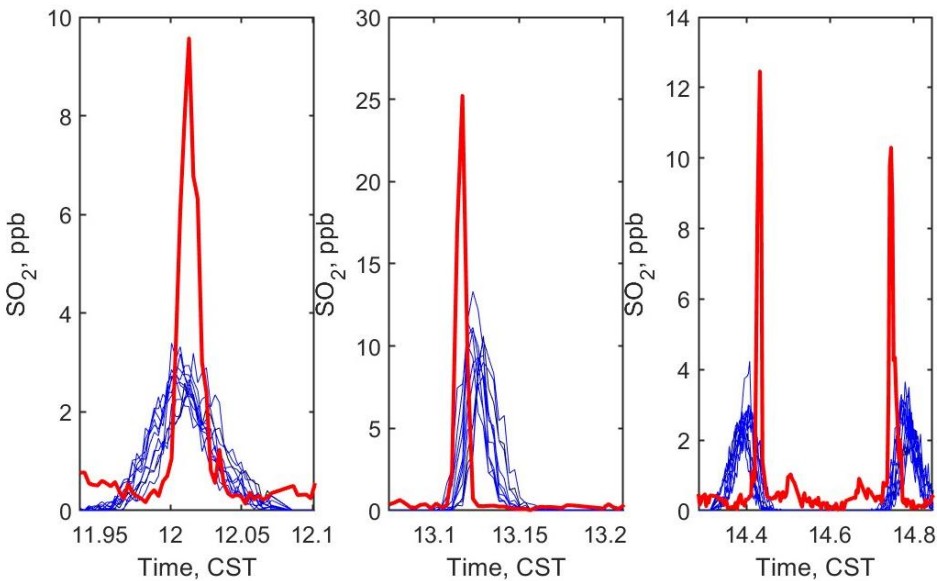

Figure 2: Time series of observed (red) and modeled (control in black, ensemble members in blue) $SO_2$ mixing ratio

along the flight tracks on 25 June 2013. The lower plots zoom in on the relevant segments of the upper time series to

show details of the plume magnitudes and positions.





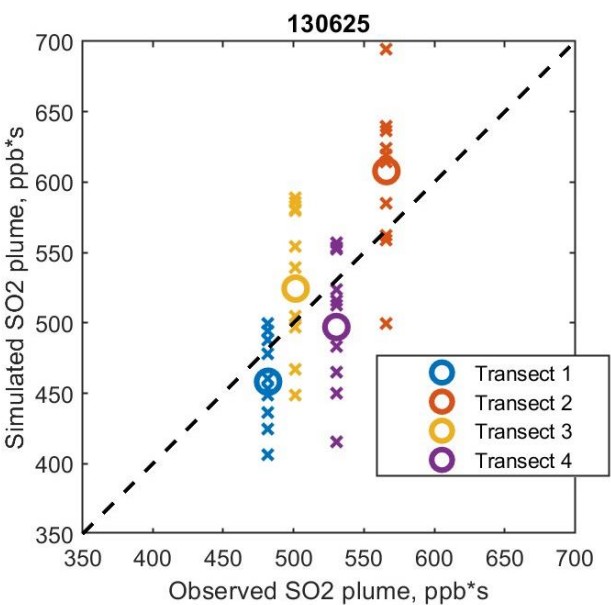

557 Figure 3: Integrated plume amounts for the four transects on 25 June 2013, comparing observations and simulations.

558 Large circles are from the control simulation, x marks are from the ensemble members. The amounts are found by

559 integrating the mixing ratio in time across each of the plumes shown in figure 2.

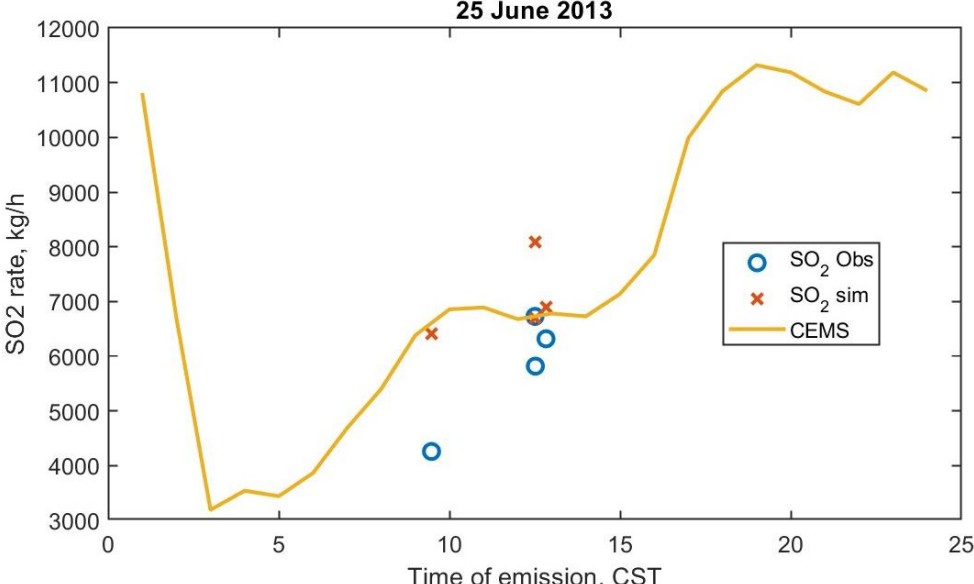

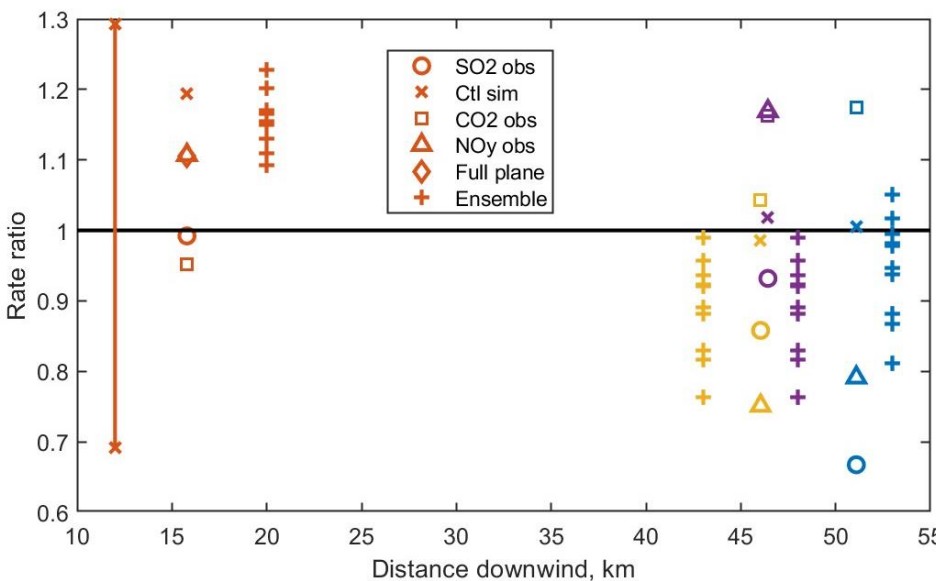

Figure 4: Emission rate estimates from mass balance for four transects on 25 June 2013. Upper panel: CEMS

emission rate through the day with observed and simulated emission estimates shown at estimated time of emission.

Lower panel: Ratio of emission rate derived from different estimates (method and species, symbols as in legend) to

CEMS emission at estimated time of emission. Explanation of legend: "obs" is mass balance using observations; "Ctl

sim" is mass balance using control simulated tracer, wind, and mixing height; "full plane" is derived by integrating the

full x-z plane in the simulation at the transect latitude; "ensemble" is mass balance using simulated tracer, wind, and



mixing height for each ensemble member.  Vertical bar is uncertainty estimate (one standard deviation, ±30%) on

observation-based mass balance for transect 2, and its length applies to all transects.  The four transects are colored

blue, red, yellow, and purple respectively. Ensemble estimates are offset slightly along the x axis for clarity.



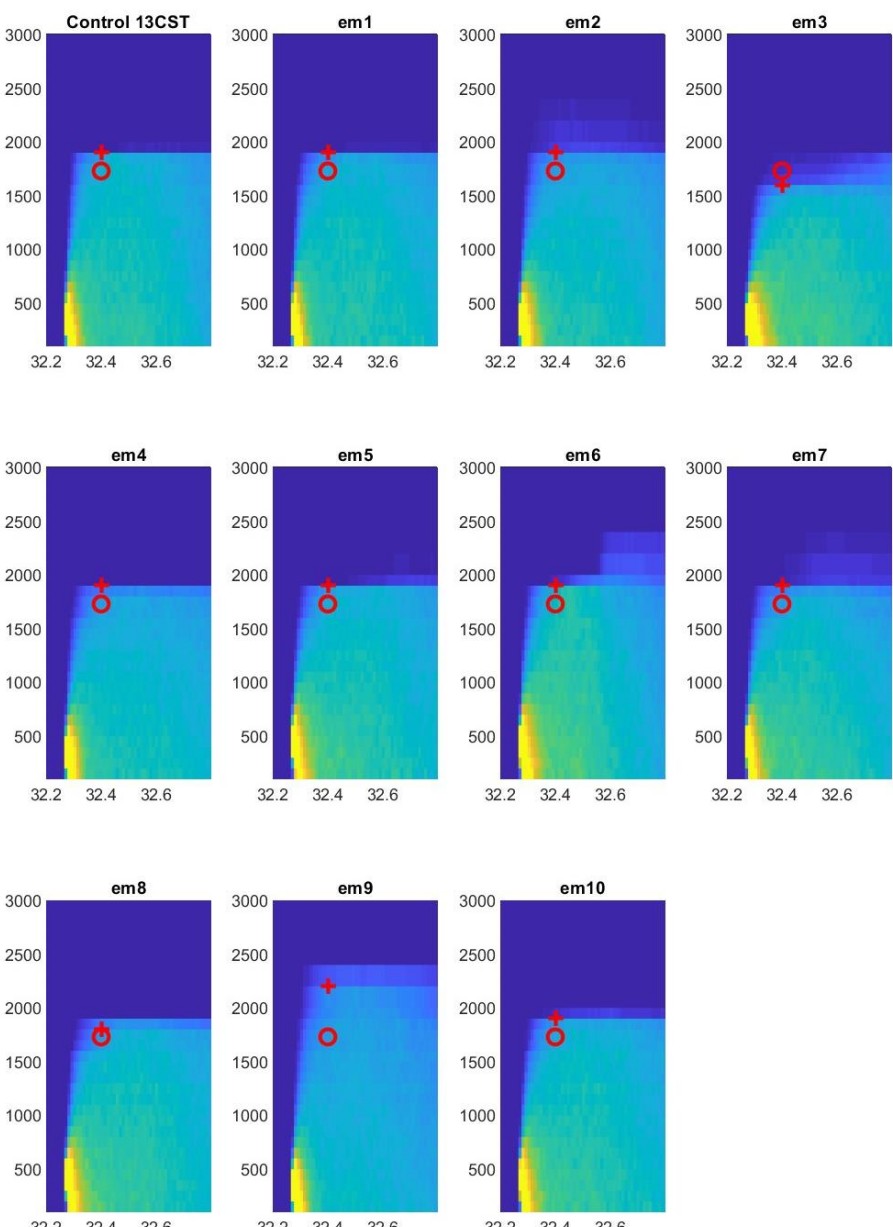

Figure 5: Latitude – height cross sections of SO$_2$ mixing ratio simulated with the control meteorology and each of the

ensemble members on 25 June 2013. The cross sections are shown at 1300 CST. The color scale is linear and is

allowed to saturate near the source. Two estimates of the mixing height are shown at the latitude of transect 2:

Observed (o, one value for all subplots), and determined from the concentration profile (+). See text for details.

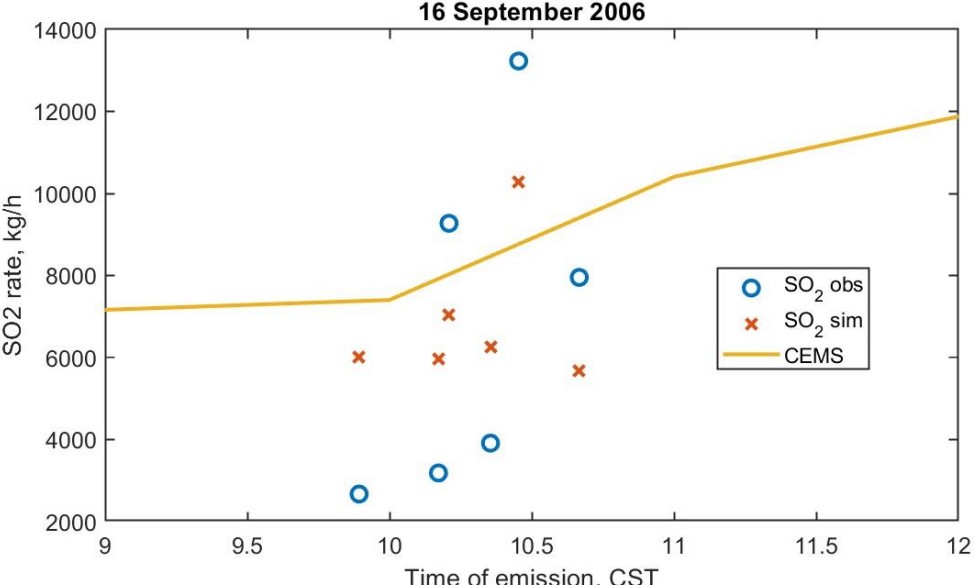

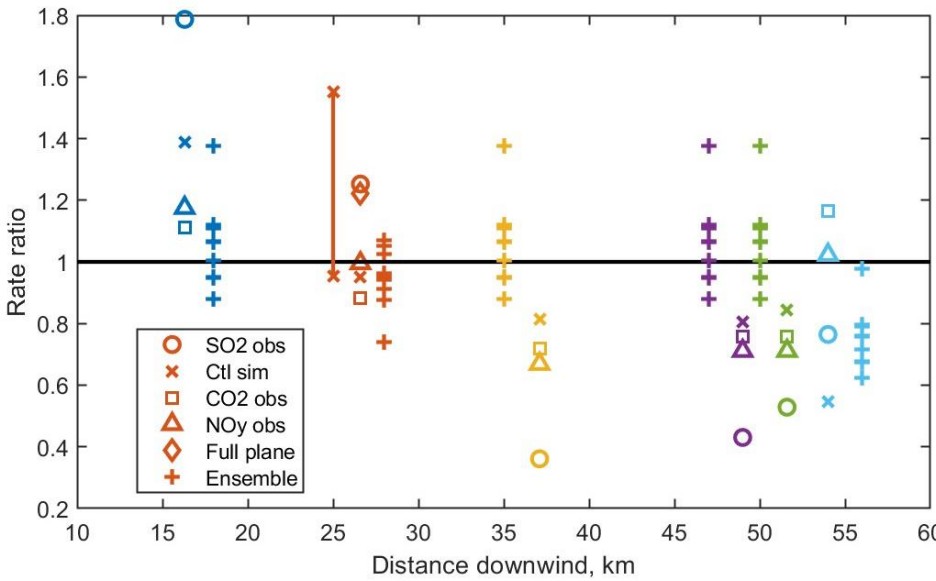

Figure 6: Emission rate estimates from mass balance for six transects on 16 September 2006. Upper panel: CEMS

SO₂ emission rate with observed and simulated emission estimates shown at estimated time of emission. Lower panel:

Ratio of emission rate derived from different estimates (method and species) to CEMS emission at estimated time of

emission. Explanation of legend: "obs" is mass balance using observations; "Ctl sim" is mass balance using control



simulated tracer, wind, and mixing height; "full plane" is derived by integrating the full x-z plane in the simulation at

the transect latitude; "Ensemble" is mass balance using simulated tracer, wind, and mixing height for each ensemble

member.  Vertical bar is uncertainty estimate (one standard deviation, ±30%) on observation-based mass balance for

transect 2. The six transects are colored blue, red, yellow, purple, green, and cyan respectively. Ensemble estimates are

offset slightly along the x axis for clarity.



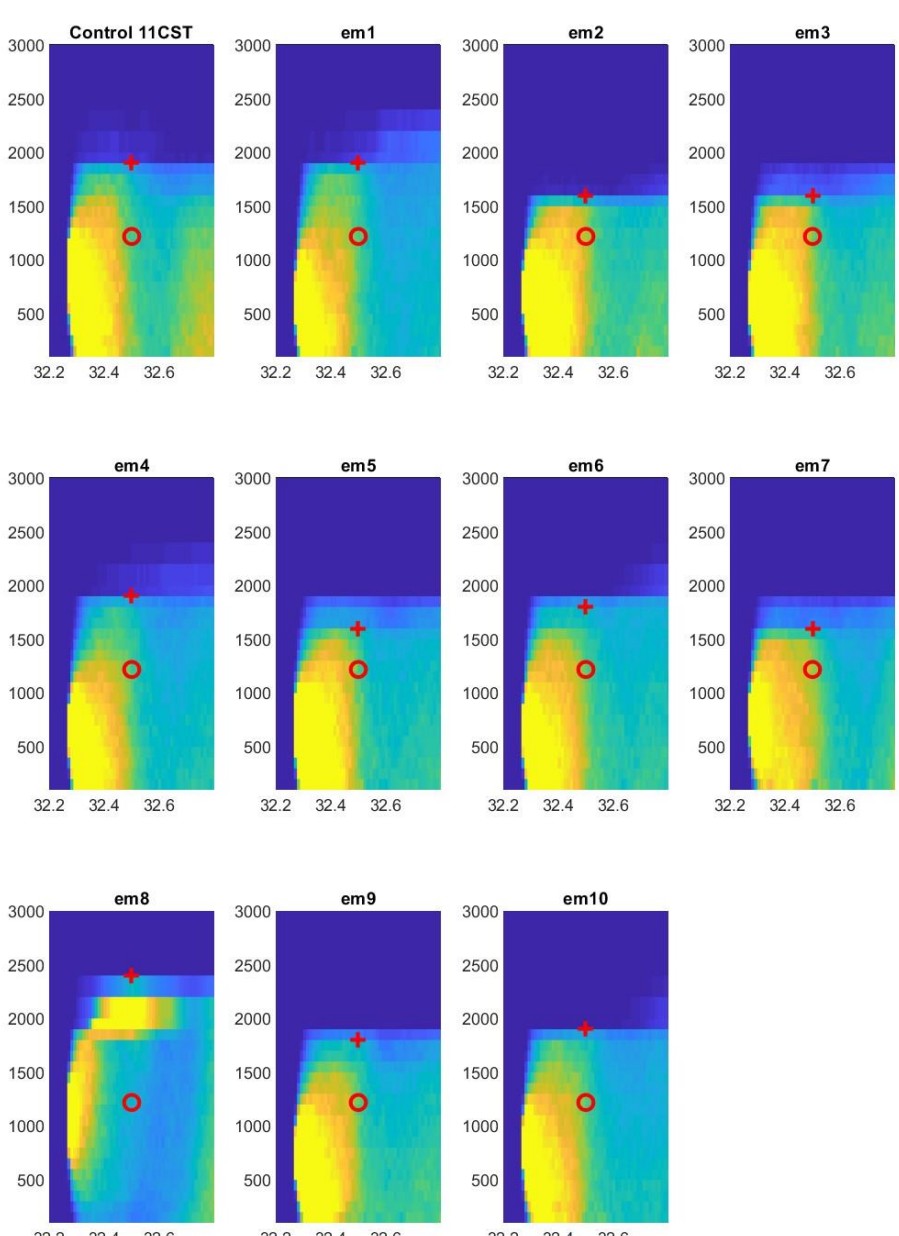

Figure 7: Latitude – height cross sections of tracer mixing ratio simulated with the control meteorology and each of the ensemble members on 16 September 2006. The cross sections are shown at 1100 CST. The color scale is linear and is allowed to saturate near the source. Two estimates of the mixing height are shown at the latitude of transect 2: Observed (o, one value for all subplots), and determined from the concentration profile (+). See text for details.

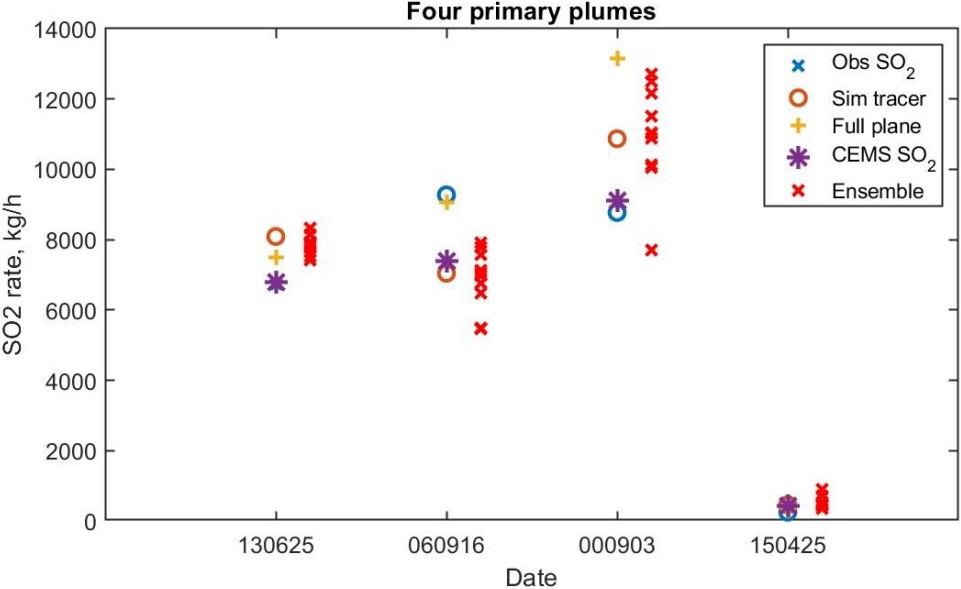

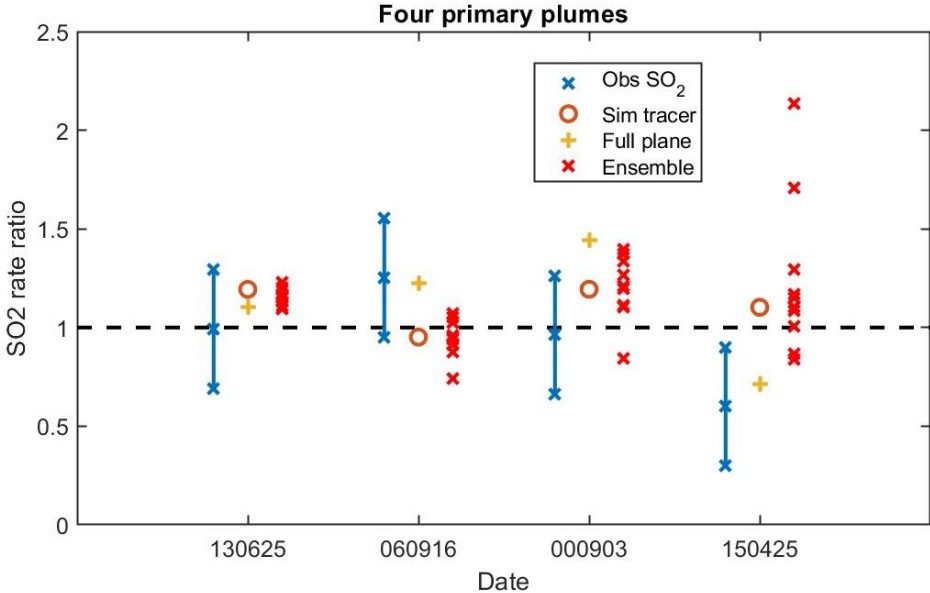

Figure 8: Emission rate estimates from mass balance for the four primary transects highlighted in table 1. Upper

panel: Absolute SO₂ emission rates. Lower panel: Ratio of SO₂ emission rates derived from different estimates

(method and species) to CEMS emission at estimated time of emission. Explanation of legend: "obs" is mass balance

using observations; "sim" is mass balance using simulated tracer, wind, and mixing height; "full plane" is derived by





integrating the full x-z plane in the simulation at the transect latitude; "Ensemble" is mass balance using simulated

tracer, wind, and mixing height for each ensemble member. Vertical bar is uncertainty estimate on observation-based

mass balance.