# Peer review of "Errors in top-down estimates of emissions using a known source"

_Atmospheric Chemistry and Physics, 2020_

## Referee Comment (RC1) · Anonymous Referee #2 · 19 May 2020

The paper presents a careful assessment of uncertainties in estimation of emissions from atmospheric plume observations. It is certainly of interest to the scientific community. In general the paper is well written, and I recommend publication after the following concerns have been addressed.

General Comments:

There are only a very few references used in the Introduction. For mass balance for example, here are two exemplary references: Karion, A., Sweeney, C., Pétron, G., Frost, G., Michael Hardesty, R., Kofler, J., Miller, B. R., Newberger, T., Wolter, S., Banta, R., Brewer, A., Dlugokencky, E., Lang, P., Montzka, S. A., Schnell, R., Tans, P., Trainer, M., Zamora, R., and Conley, S.: Methane emissions estimate from airborne measurements over a western United States natural gas field, Geophysical Research Letters,

40, 4393-4397, doi:10.1002/grl.50811, 2013. Turnbull, J. C., Karion, A., Fischer, M. L., Faloona, I., Guilderson, T., Lehman, S. J., Miller, B. R., Miller, J. B., Montzka, S., Sherwood, T., Saripalli, S., Sweeney, C., and Tans, P. P.: Assessment of fossil fuel carbon dioxide and other anthropogenic trace gas emissions from airborne measurements over Sacramento, California in spring 2009, Atmos. Chem. Phys., 11, 705-721, 10.5194/acp-11-705-2011, 2011. Also, the sections on errors, uncertainty, and ensemble forecasting could benefit from some key references. Furthermore, some references should be used in the discussion of the mixing height and well-mixedness (lines 386 – 395).

Specific comments

Fig. 1 missing? The Fig. 1 caption does not quite fit to any of the figures I can see in the manuscript. indicates "The x marks the power plant location", but I can't find the figure with an "x". A map showing the power plant location as well as the locations/directions of the transects and the orientation of the cross sections shown in Fig. 5 is certainly required.

Fig. 4, bottom panel: The figure is not very clear. Each transect has a specific colour, and ensemble estimates are offset in x-value, but e.g. for the 2nd transect there are three different x-values. For clarity it would be better to reduce the x-offset between estimates within each transect, and to mention in the caption that also the error estimate for the observed mass balance is shown for the 2nd transect.

Fig. 5: It would be informative to also show the simulated mixing height as function of latitude, and the exact location of the different transects (also for Fig. 7).

Line 254: "along-wind direction" is this the wind direction at the source, at the measurement location, or is this along a mean wind trajectory?

Fig. 7: Most cross sections show minima in mixing ratios at a latitude of around 32.6. Is this due to changes in wind speed or wind direction? Maybe a figure showing a map of

[Figure]

a horizontal cross section (lat-lon) for one of the simulated plumes at different heights would help.

Line 293: Do the ERA5 mixing heights show a significant temporal change?

Line 391: Both, potential temperature and water vapour mixing ratio show gradients across the top of the PBL (although of opposite sign), so why should one quantity be well mixed in the PBL and not the other? A reference would be needed here.

Line 410: "esimate" > "estimate"

Line 410: It would be interesting to examine the correlation (positive or negative) between wind speed and mixing height in the ensemble fields, this would help clarifying the issue of compensating errors.

---

## Referee Comment (RC2) · Anonymous Referee #3 · 20 Jul 2020

**1  General comments**

The paper address important topic with interesting data. To quantify uncertainties is definitely important topic in atmospheric science today. The paper is well written and clear to follow, however, some clarification would be beneficial, see specific and technical comments bellow.

[Figure]

**2  Specific comments**

1. I suggest the authors to divide some sections to subsections in order to increase clarity and readeability of the paper. E.g., list of errors in Discussion could be considered in separate subsections. Similarly the Methods section etc.

2. It would be definitelly worth to add a subsection with mass-balance method describtion since it is the core methodology of the paper. It is in the paper already, but some part is in Introduction, some in Methods, please, consider to consolidate it.

3. On line 151 (and also elsewhere), the authors stated that "A robust estimate of the background (concentration not attributable to the source of interest) is required." Could you please elaborate in more details the estimation of background in your case?

**3  Technical corrections**

1. please, follow the ACP house standards, see https://www.atmospheric-chemistry-and-physics.net/for_authors/manuscript_preparation.html. Specifically, figures should be references as "Fig. X", tables as "Table X", etc.

2. In multipanel figures, please, use (a), (b), etc. for each panel according to house standard

---

## Author Comment (AC1) · 21 Aug 2020

Author's response to reviewer comments

Reviewer comments in *italic*, responses in normal font

**Referee #2**

*The paper presents a careful assessment of uncertainties in estimation of emissions from atmospheric plume observations. It is certainly of interest to the scientific community. In general the paper is well written, and I recommend publication after the following concerns have been addressed.*

Thanks for understanding what we are trying to accomplish. A "careful assessment" is precisely the point.

*General Comments:*

*There are only a very few references used in the Introduction. For mass balance for example,here are two exemplary references…*

We added the suggested Karion et al. and Turnbull et al. references, as well as Karion et al. 2015.

*Also, the sections on errors, uncertainty, and ensemble forecasting could benefit from some key references. Furthermore, some references should be used in the discussion of the mixing height and well-mixedness (lines 386 – 395).*

A few key references have been added to the specified sections.

*Specific comments*

*Fig. 1 missing? The Fig. 1 caption does not quite fit to any of the figures I can see in the manuscript. indicates "The x marks the power plant location", but I can't find the figure with an "x". A map showing the power plant location as well as the locations/directions of the transects and the orientation of the cross sections shown in Fig. 5 is certainly required.*

Figure 1 is correct in the posted version of the discussion paper. Was the reviewer looking at the first (pre-posting) version, which was missing figure 1?

*Fig. 4, bottom panel: The figure is not very clear. Each transect has a specific colour, and ensemble estimates are offset in x-value, but e.g. for the 2nd transect there are three different x-values. For clarity it would be better to reduce the x-offset between estimates within each transect, and to mention in the caption that also the error estimate for the observed mass balance is shown for the 2nd transect.*

The offset has been reduced for transect 2. The caption already mentions that the error bar applies to the 2$^{nd}$ transect, but because the figure shows ratios, the length of the bar applies for all transects.

*Fig. 5: It would be informative to also show the simulated mixing height as function of latitude, and the exact location of the different transects (also for Fig. 7).*

We think the figure is complex enough without adding further lines and symbols. Transect 2 is the one that is primarily in the text, and the point that mixing height and degree of mixing vary among the ensemble members is made with the existing figure.

*Line 254: "along-wind direction" is this the wind direction at the source, at the measurement location, or is this along a mean wind trajectory?*

It is the approximate wind direction, in fact just south-north, which is within 10 degrees of the measured winds on the aircraft track. This has been clarified in the text.

*Fig. 7: Most cross sections show minima in mixing ratios at a latitude of around 32.6. Is this due to changes in wind speed or wind direction? Maybe a figure showing a map of a horizontal cross section (lat-lon) for one of the simulated plumes at different heights would help.*

We hadn't noticed this minimum. It must be due to changes in wind speed through the morning hours. Horizontal cross-sections are not enlightening because the plume spreads out so much in the horizontal that the subtle minumum cannot be seen.

*Line 293: Do the ERA5 mixing heights show a significant temporal change?*

There is not much change in the ERA5 mixing heights during the flight times.

*Line 391: Both, potential temperature and water vapour mixing ratio show gradients across the top of the PBL (although of opposite sign), so why should one quantity be well mixed in the PBL and not the other? A reference would be needed here.*

It is precisely because the gradients are of opposite sign. For heat, both surface and entrainment fluxes warm the PBL. For moisture, surface fluxes moisten but entrainment fluxes dry, so the well-mixed state is harder to achieve. This is well known but not well published. We have included the Gao et al. 2018 reference for those who wish to explore further.

*Line 410: "esimate" > "estimate"*

Corrected.

*Line 410: It would be interesting to examine the correlation (positive or negative) between wind speed and mixing height in the ensemble fields, this would help clarifying the issue of compensating errors.*

We were also curious about this, but found no significant correlations (plus or minus 0.25 or less except 150425, when it was 0.48).

**Referee #3:**

*1 General comments*

*The paper address important topic with interesting data. To quantify uncertainties is definitely important topic in atmospheric science today. The paper is well written and clear to follow, however, some clarification would be beneficial, see specific and technical comments bellow.*

Thanks for the appreciative comment.

*2 Specific comments*

*1. I suggest the authors to divide some sections to subsections in order to increase clarity and readeability of the paper. E.g., list of errors in Discussion could be considered in separate subsections. Similarly the Methods section etc.*

We have divided the Methods section into two subsections, Mass Balance and Models. Similarly, the Discussion section has been divided into two subsections.

*2. It would be definitelly worth to add a subsection with mass-balance method describtion since it is the core methodology of the paper. It is in the paper already, but some part is in Introduction, some in Methods, please, consider to consolidate it.*

The mass balance description has been lengthened considerably and consolidated into the Methods section.

*3. On line 151 (and also elsewhere), the authors stated that "A robust estimate of the background (concentration not attributable to the source of interest) is required." Could you please elaborate in more details the estimation of background in your case?*

We have added text describing the background determination in the Methods section.

*3 Technical corrections*

*1. please, follow the ACP house standards, see https://www.atmosphericchemistry-and-physics.net/for_authors/manuscript_preparation.html. Specifically, figures should be references as "Fig. X", tables as "Table X", etc.*

Done.

*2. In multipanel figures, please, use (a), (b), etc. for each panel according to house standard*

Done.